# Learning to Repair Software Vulnerabilities with Generative Adversarial Networks

**Jacob A. Harer**[1,2], **Onur Ozdemir**[1], **Tomo Lazovich**[3*], **Christopher P. Reale**[1],
**Rebecca L. Russell**[1], **Louis Y. Kim**[1], **Peter Chin**[2]

[1]Draper, Cambridge, MA
[2]Department of Computer Science, Boston University, Boston, MA
[3]Lightmatter, Boston, MA

{jharer,oozdemir,creale,rrussell,lkim}@draper.com,
tomo@lightmatter.ai, spchin@cs.bu.edu

## Abstract

Motivated by the problem of automated repair of software vulnerabilities, we propose an adversarial learning approach that maps from one discrete source domain to another target domain without requiring paired labeled examples or source and target domains to be bijections. We demonstrate that the proposed adversarial learning approach is an effective technique for repairing software vulnerabilities, performing close to seq2seq approaches that require labeled pairs. The proposed Generative Adversarial Network approach is application-agnostic in that it can be applied to other problems similar to code repair, such as grammar correction or sentiment translation.

## 1 Introduction

Security vulnerabilities in software programs pose serious risks to computer systems. Malicious users can compromise programs through their vulnerabilities to force them to behave in undesirable ways (e.g. crash, expose sensitive user information, etc.). Thousands of such vulnerabilities are reported publicly to the Common Vulnerabilities and Exposures database (CVE) each year, and many more are discovered internally in proprietary code and patched [1, 2]. These vulnerabilities are often the result of errors made by programmers, and, due to the prevalence of open source software and code re-use, can propagate quickly.

In this paper, we address the problem of learning to automatically repair the source code of software containing security vulnerabilities. This problem is analogous to grammatical error correction, in which a grammatically incorrect sentence is translated into a correct one. In our case, bad source code (that contains a vulnerability) takes the place of an incorrect sentence and is repaired into good source code.

Neural Machine Translation (NMT) systems have recently achieved the state-of-the-art performance on language translation and correction tasks [3, 4, 5, 6]. These models use an encoder-decoder approach to transform an input sequence $\mathbf{x} = (x_0, x_1...x_T)$ into an output sequence $\mathbf{y} = (y_0, y_1...y_{T'})$, e.g., translating a sequence of words forming a sentence in English to one in German. By far the most common method of training NMT systems is to use labeled pairs of examples to compare the likelihood of network output to a desired version, necessitating a one-to-one mapping between input and desired output data. This can be difficult to obtain as in most cases it requires costly hand annotation.

In many sequence-to-sequence (seq2seq) applications, it is much easier to obtain unpaired data, i.e., data from both source and target domains without any matching pairs, since this

only requires data to be labeled as either source or target. For example, in natural language translation it is easy to obtain monolingual corpora in different languages with almost no cost. For source code, automated error detection methods exist, such as static analyzers or machine learning approaches, which can be used to label code as having vulnerabilities or not, but do not provide one-to-one pairing between labeled sets [7, 8].

Our approach to address this problem is adversarial learning with Generative Adversarial Networks (GANs) [9]. This approach allows us to train without paired examples. We employ a traditional NMT model as the generator, and replace the typical negative likelihood loss with the gradient stemming from the loss of an adversarial discriminator. The discriminator is trained to distinguish between NMT-generated outputs and real examples of desired output, and so its loss serves as a proxy for the discrepancy between the generated and real distributions. This problem has three main difficulties. Firstly, sampling from the output of NMT systems, in order to produce discrete outputs, is non-differentiable. We address this problem by using a discriminator which operates directly on the expected (soft) outputs of the NMT system during training, which we thoroughly discuss in Section 3.2. Secondly, adversarial training does not guarantee that the generated code will correspond to the input bad code (i.e. the generator is trained to match distributions, not samples). To enforce the generator to generate useful repairs, (i.e., generated code is a repaired version of input bad code), we condition our NMT generator on the input **x** by incorporating two novel generator loss functions, described in Section 3.3. Thirdly, the domains we consider are not bijective, i.e., a bad code can have more than one repair or a good code can be broken in more than one way. The regularizers we propose in Section 3.3 still work in this case. We should note that although our motivation is to repair source code, the approach and the techniques proposed in this paper are application-agnostic in that they can be applied to other similar problems, such as correcting grammar errors or converting between negative and positive sentiments (e.g., in online reviews.). Additionally, while software vulnerability repair is a harder problem than detection, our proposed repair technique can leverage the same datasets used for detection and yields a much more explainable and useful tool than detection alone.

## 2    Related Work

### 2.1    Software Repair

Much research has been done on automatic repair of software. Here we describe previous data-driven approaches (see [10] for a more extensive review of the subject). Two successful recent approaches are that of Le et al. [11] and Long and Rinard [12]. Le et al. mine a history of bug fixes across multiple projects and attempt to reuse common bug fix patterns on newly discovered bugs. Long and Rinard learn and use a probabilistic model to rank potential fixes for defective code. These works, along with the majority of past work in this area, require a set of test cases which is used to rank and validate produced repairs. Devlin et al. [13] avoid the need for test cases by generating repairs with a rule based method and then ranking them using a neural network. Gupta et al. [14] take this one step further by training a seq2seq model to directly generate repairs for incorrect code. Hence, the work in [14] most closely resembles our work, but has the major drawback of requiring paired training data.

### 2.2    GANs

GANs were first introduced by Goodfellow et al. [15] to learn a generative model of natural images. Since then, many variants of GANs have been created and applied to the image domain [16, 17, 18, 19, 20]. GANs have generally focused on images due to the abundance of data and their continuous nature. Applying GANs to discrete data (e.g. text) poses technically challenging issues not present in the continous case (e.g. propagating gradients through discrete values). One successful approach is that of Yu et al. [21], which treats the output of the discriminator as a reward in a reinforcement learning setting. This allows the sampling of outputs from the generator since gradients do not need to be passed back through the discriminator. However, since a reward is provided for the entire sequence, gradients computed for the generator do not provide information on which parts of the

output sequence the discriminator thinks is incorrect, resulting in long convergence times. Several other approaches have had success with directly applying an adversarial discriminator to the output of a sequence generator with likelihood output. Zhang et al. [22] replace the traditional GAN loss in the discriminator with a Maximum Mean Discrepancy (MMD) metric in order to stabilize GAN training. Both Press et al. [23] and Rajeswar et al. [24] are able to generate fairly realistic looking sentences of modest length using Wasserstein GAN [17], which is the approach we adopt in this paper.

Work has also been done on how to condition a GAN's generator on an input sequence $\mathbf{x}$ instead of a random variable. This can easily be performed when paired data is available, by providing the discriminator with both $\mathbf{x}$ and $\mathbf{y}$, thereby formulating the problem as in the conditional approach of Mirza and Osindero [25, 26]. This approach, however, is clearly more difficult when pairs are not available. One approach is to enforce conditionality through the use of dual generator pairs which translate between domains in opposite directions. For example, Gomez et al. apply the cycle GAN [27] approach to cipher cracking [28]. They train two generators, one to take raw text and produced ciphered text, and the other to undo the cipher. Having two generators allows Gomez et al. to encrypt raw data using the first generator, then decrypt it with the other, ensuring conditionality by adding a loss function which compares this doubly translated output with the original raw input. Lample et al. [29] adopt a somewhat similar approach for NMT. They translate using two encoder/decoder pairs which convert from a given language to a latent representation and back respectively. They then use an adversarial loss to ensure that the latent representations are the same between both languages, thus allowing translation by encoding from one language and then decoding into the second. For conditionality they adopt a similar approach to Gomez et al. by fully translating a sentence from one language to another, translating it back, and then comparing the original sentence to the produced double translation.

The approaches of both Gomez et al. and Lample et al. rely on the ability to transform a sentence across domains in both directions. This makes sense in many translation spaces as there are a finite number of reasonable ways to transform a sentence in one language to a correct one in the other. This allows for a network which finds a single mapping from every point in one domain to a single point in the other domain, to still cover the majority of translations. Unfortunately, in a sequence correction task such as our problem, one domain contains all correct sequences, while the other contains everything not in the correct domain. Therefore, the mapping from correct to incorrect is not one-to-one, it is one to many. A single mapping discovered by a network would fail to elaborate the space of all bad functions, thus enforcing conditionality only on the relatively small set of bad functions it covers. Therefore, we propose to enforce conditionality using a self-regularization term on the generator, similar in nature to that used by Shrivastava et al. [30] to generate realistic looking images from simulated ones.

## 3    Formulation

GANs are generative models originally proposed to generate realistic images, $y$, from random noise vectors, $z$ [9]. GANs find a mapping $G : z \rightarrow y$ by framing the learning problem as a two player minimax game between a generator $G(\cdot)$ and a discriminator $D(\cdot)$, where the generator learns to generate realistic looking data samples by minimizing the performance of a discriminator whose goal is to maximize its own performance on discriminating between generated and real samples.

Our problem in this paper is different from the original GAN problem in that our goal is to find a mapping between two discrete valued domains, namely between a given bad code (or source) domain $X$ and a good code (or target) domain $Y$ by using unpaired training samples $\{x_i\}_{i=1}^N$ and $\{y_i\}_{i=1}^M$, where $x_i \in X$ and $y_i \in Y$.

### 3.1    Adversarial Loss

The original GAN loss of Goodfellow et al. [9] is expressed as

$$\mathcal{L}_{GAN}(D,G) = \mathbb{E}_{y \sim P(y)}[\log\ D(y)] + \mathbb{E}_{x \sim P(x)}[\log(1\ -\ D(G(x))] \tag{1}$$

where the optimal generator is $G^* = \arg\min_G \max_D \mathcal{L}_{GAN}(D,G)$. It is well known that this loss can be unstable when the support of the distributions of generated and real samples do not overlap [16]. This causes the discriminator to provide zero gradients. Further, this standard loss function can lead to mode collapse, where the resulting samples come from a single mode of the real data distribution. To alleviate these problems, Arjovsky et al. [17] proposed the Wasserstein GAN (WGAN) loss which instead uses the Wasserstein-1 or Earth-Movers (EM) distance between generated and real data samples in the discriminator. EM distance is relatively straightforward to estimate and leads to the easily computable loss function:

$$\mathcal{L}_{WGAN}(D,G) = \mathbb{E}_{y \sim P(y)}[D(y)] - \mathbb{E}_{x \sim P(x)}[D(G(x))] \tag{2}$$

where the discriminator function $D$ is constrained to be 1-Lipschitz. We use WGAN in our model as it leads to more stable training.

## 3.2   GANs with Discrete Data

One of the main challenges of adversarial training with discrete sequences is that sampling from the output of NMT systems in order to produce discrete outputs is non-differentiable. The goal of training is to generate samples from the unknown distribution of real sequences $P_Y$, which can be factorized as

$$P_Y(\mathbf{y}) = P(y_0) \prod_{t=1}^{T} P(y_t | y_0, ... y_{t-1}) \tag{3}$$

where each conditional distribution $P(y_t | y_0, ... y_{t-1})$ is estimated (using an RNN generator in our case) with a softmax output

$$\hat{P}(y_t | y_0 ... y_{t-1}) = \mathbf{s}_t \triangleq \text{softmax}(f(y_{t-1}, \mathbf{h}_{t-1})) \tag{4}$$

where $f(\cdot)$ and $\mathbf{h}_t$ denote the generator network and the hidden state of the RNN at time $t$, respectively. Ideally, we would sample from $\mathbf{s}_t$ to generate a sequence and provide that to the discriminator along with the real data for training, but this sampling process is non-differentiable. Instead, we provide the discriminator with $\mathbf{s}_t$ directly. Since each $\mathbf{s}_t$ is dependent on the previously produced output token and the RNN state, we still need to sample $y_{t-1}$ from $\mathbf{s}_{t-1}$ using $\arg\max$ to generate $\mathbf{s}_t$. Note that $\mathbf{s}_t$ can be interpreted as the *soft* one-hot representation as it corresponds to the expectation of one-hot vectors with respect to the conditional distribution in (4). Although this soft representation alleviates the issue of non-differentiability, it may introduce potential issues with the discriminator which we discuss next.

Note that since each generator output $\mathbf{s}_t$ is a probability vector it will almost surely not be a one hot vector. In other words, while every real token, $y_t$, lies on one of the vertices of the standard $V-1$ dimensional simplex, our generated outputs, $\mathbf{s}_t$, lie on the interior of the simplex. This implies that $P_r$ and $P_s$ have disjoint supports and are perfectly separable in theory. Therefore, there exists a 'trivial' discriminator which looks at each token independently and discriminates based on whether a sequence consists of one-hot vectors or not. Such a discriminator would not provide useful information for training the generator since it does not pay attention to the sequential dependencies between tokens. Nevertheless, we conjecture that simple discriminator architectures do not have this problem, since such a 'trivial' discriminator may be hard to realize in practice. This was verified in our experiments where we found that relatively shallow networks, such as those using only a single convolutional layer, performed better than deeper ones.

There is related work in the literature [23, 24, 28] that reported avoiding this 'trivial' discriminator by using the improved Wasserstein GAN (WGAN-GP) loss [31]. However, in our implementations, we had more success with the original version of the Wasserstein GAN, which uses clipped weights in the discriminator (after both versions had sufficient hyper-parameter tuning). We believe that this is due to weight clipping in the original Wasserstein GAN that forces the discriminator to learn simpler functions, as was shown in the improved WGAN paper [31]. These simpler functions do not allow the discriminator to simply focus on one-hot vectors and force it to pay attention to sequential dependencies between tokens. To further analyze this point, we provide some visualizations in Figure

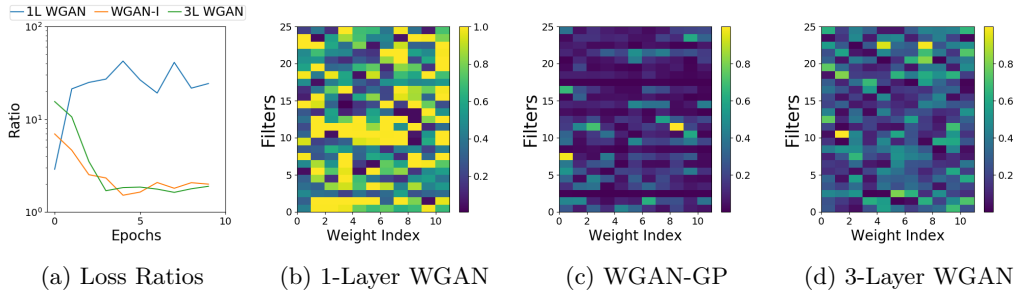

| (a) Loss Ratios | (b) 1-Layer WGAN | (c) WGAN-GP | (d) 3-Layer WGAN |

Figure 1: (a) Wasserstein loss ratios between correctly and incorrectly generated pairs during training. (b-d) Weights of 1-layer 1D CNN with WGAN loss, 1-layer 1D CNN with WGAN-GP loss, and 3-layer 1D CNN with WGAN, respectively.

1, where we use a paired dataset for analysis purposes. We sample a random set of data pairs, where $x$ is a bad version of $y$, and compute Wasserstein loss values, $\mathcal{L}_{WGAN}(D, G)$ as defined in (2), for two separate cases. For the first loss calculation, we select pairs where the generator $G(x)$ generates correct outputs $(G(x) = y)$, and for the second loss, the generated outputs are incorrect $(G(x) \neq y)$. We then take the ratio of these two loss values and plot them in Figure 1a for three different discriminator settings, namely i) 1-layer CNN with WGAN loss; ii) 1-layer CNN with WGAN-GP loss; and 3) 3-layer CNN with WGAN loss. A discriminator which only differentiates inputs based on whether they are one-hot vectors or not should have very similar loss values for the two cases resulting in a loss ratio of $\sim 1$, since in neither case does the generator produce one-hot vectors. As we observe in Figure 1a, the simpler network architecture (1-layer CNN in this case) with the original Wasserstein loss provides better separation, i.e., better signal, for training the generator. This is further emphasized by Figures 1b-1d where we show normalized weights of the 1-D convolutional filters (whose kernel size is 11) on the first convolutional layer in each network. Filters for the simplest network in Figure 1b have a low degree of sparsity, implying that they are aggregating data from multiple tokens taking into account sequential dependencies, whereas the networks in both Figures 1c and 1d have a much higher degree of sparsity, often emphasizing only a single token at a time, which we would expect for discriminators paying attention to individual tokens to decide based on whether a given token is one-hot or not. These observations imply an inherent trade-off. An overly complex discriminator can learn to discriminate based on spurious features, i.e., whether a vector is one-hot or not, which can lead to overfitting. On the other hand, a very simple discriminator will not accurately model the data and, therefore, not provide any useful information to the generator. One needs to treat this trade-off as one would treat a hyperparameter, by tuning the discriminator model on an application by application basis.

We should also mention that there are two other approaches proposed in the literature to overcome the issues we discussed above. The first approach is to (linearly and deterministically) embed each one-hot vector into a lower dimensional space [23]. This approach is still vulnerable to the problem of a sufficiently complex discriminator ignoring sequential dependencies since these embedding are deterministic. We found this to be the case in practice as well; adding an embedding the the discriminator alone produced no noticeable improvement and still required the use of simple networks. The second alternative approach is to reparamaterize the discrete sampling process via a continuous relaxation using the Gumbel-softmax distribution [32, 33]. This approach, due to continuous relaxation, still generates (random) outputs via a softmax function, which are therefore similar to our soft one-hot outputs. We experimented with this approach and did not observe any improvements.

### 3.3 Domain Mapping with Self-Regularization

In the context of source code repair, or more generally sequence correction, we need to constrain our generated samples $y$ to be a corrected versions of $x$. Therefore, we have the following two requirements: (1) correct sequences should remain unchanged when passed through the generator; and (2) repaired sequences should be close to the original corresponding incorrect input sequences.

We explore two regularizers to address these requirements. As our first regularizer, in addition to GAN training, we train our generator as an autoencoder on data sampled from correct sequences. This directly enforces item (1), while indirectly enforcing item (2) since the autoencoder loss encourages subsequences which are correct to remain unchanged. The autoencoder regularizer is given as

$$\mathcal{L}_{AUTO}(G) = \mathbb{E}_{x \sim P(x)} \left[ -x \log(G(x)) \right]. \tag{5}$$

As our second regularizer, we enforce that the frequency of each token in the generated output remains close to the frequency of the input tokens. This enforces item (2) with the exception that it may allow changes in the order of the sequence, e.g., arbitrary reordering does not increase this loss. However, the GAN loss alleviates this issue since arbitrary reordering produces incorrect sequences which differ heavily from $P(\mathbf{y})$. Our second regularizer is given as

$$\mathcal{L}_{FREQ}(G) = \mathbb{E}_{x \sim P(x)} \left[ \sum_{i=0}^{n} \|\mathrm{freq}(x, i) - \mathrm{freq}(G(x), i)\|_2^2 \right]. \tag{6}$$

where $n$ is the size of the vocabulary and $\mathrm{freq}(x, i)$ is the frequency of the i$^{\mathrm{th}}$ token in x.

## 4   Putting It All Together - Proposed GAN Framework

The generator in our network consists of a standard NMT system with an attention mechanism similar to that of Luong et al [34]. For all experiments the encoder and decoder consist of multi-layer RNNs utilizing Long Short-Term Memory (LSTM) units [35]. We use a dot-product attention mechanism as per [34]. We use convolution based discriminators since they have been shown to be easier to train and to generally perform better than RNN based discriminators [26]. Additional network details are provided in the Supplementary Material.

We have two different regularized loss models given as

$$\mathcal{L}(D, G) = \mathcal{L}_{WGAN}(D, G) + \lambda \mathcal{L}_{AUTO}(G) \tag{7}$$
$$\mathcal{L}(D, G) = \mathcal{L}_{WGAN}(D, G) + \lambda \mathcal{L}_{FREQ}(G) \tag{8}$$

where $\mathcal{L}_{AUTO}(G)$ and $\mathcal{L}_{FREQ}(G)$ are defined in Section 3.3. We also experiment with the unregularized base loss model where we set $\lambda = 0$.

### 4.1   Autoencoder Pre-Training

We rely heavily on pre-training to give our GAN a good starting point. Our generators are pre-trained as de-noising autoencoders on the desired data [36]. Specifically we train the generator with the loss function:

$$\mathcal{L}_{AUTO\_PRE}(G) = \mathbb{E}_{y \sim P(y)} \left[ -y \log(G(\hat{y})) \right] \tag{9}$$

where $\hat{y}$ is the noisy version of the input created by dropping tokens in $y$ with probability 0.2 and randomly inserting and deleting $n$ tokens, where $n$ is 0.03 times the sequence length. These numbers were selected based on hyperparameter tuning.

### 4.2   Curriculum Learning

Likelihood based methods for training seq2seq networks often utilize teacher forcing during training where the input to the decoder is forced to be the desired value regardless of what was generated at the previous time step [37]. This allows stable training of very long sequence lengths even at the start of training. Adversarial methods cannot use teacher forcing since the desired sequence is unknown, and must therefore always pass a sample of $s_{t-1}$ as the input to time $t$. This can lead to unstable training since errors early in the output will be propagated forward, potentially creating gibberish in the latter parts of the sequence. To avoid this problem we adopt a curriculum learning strategy where we incrementally increase the length of produced sequences throughout training. Instead of selecting subsets of the data for curriculum training, we clip all sequences to have a predefined maximum length for

each curriculum step. Although this approach relies on the discriminator being able to handle incomplete sentences, it does not degrade the performance as long as the discriminator is briefly retrained after each curriculum update.

## 5    Experiments

GAN methods have often been criticized for their lack of easy evaluation metrics. Therefore, we focus our experiments on datasets which contain paired examples. This enables us to meaningfully evaluate the performance of our approach, even though our GAN approach does not require pairs to train. These datasets also allow us to train seq2seq networks and use their performance as an upper bound to our GAN based approach. We start our experiments by exploring two hand-curated datasets, namely sequences of sorted numbers and Context Free Grammar (CFG), which help highlight the benefits of our proposed GAN approach to address the domain mapping problem. We then investigate the harder problem of correcting errors in C/C++ code. All of our results are given in Table 1.

### 5.1    Sorting

In order to show the necessity of enforcing accurate domain mapping we generate a dataset where the repair task is to sort the input into ascending order. We generate sequences of 20 randomly selected integers (without replacement) between 0 and 50 in ascending order. We then inject errors by swapping $n$ selected tokens which are next to each other, where n is a (rounded) Gaussian random variable with mean 8 and standard deviation 4. The task is then to sort the sequence back into its original ascending order given the error injected sequence. This scheme of data generation allows us to maintain pairs of good (before error injection) and bad (after error injection) data, and to compute the accuracy with which our GAN is able to restore the good sequences from the bad. We refer to this accuracy as 'Sequence Accuracy' (or Seq. Acc.). In order to assess our domain mapping approach and evaluate the usefulness of our self-regularizer loss functions defined in Section 3.3, we also compute the percentage of sequences which have valid orderings but not necessarily valid domain mappings, which we refer to as 'Order Accuracy' (or Order Acc.).

It is clear from the results in Table 1 that the vanilla (base) GAN easily learns to generate sequences with valid ordering, without necessarily paying attention to the input sequence. This leads to high Order Accuracy, but low Sequence Accuracy. However, adding Auto or Freq loss regularizers, as in (7) and (8), significantly improves the Seq. Acc., which shows that these losses do effectively enforce correct mapping between source and target domains.

### 5.2    Simple Grammar

For our second experiment, we generate data from a simple Context Free Grammar similar to that used by Rajeswar et al. [24]. The specifics of the CFG is provided in the Supplementary Material. Our good data is selected randomly from the set of all sequences which satisfy the grammar and are less than length 20. We then inject errors into each sequence, where the number of errors is chosen as a Gaussian random variable (zero thresholded and rounded) with mean 5 and standard deviation 2. Each error is then randomly chosen to be either a deletion of a random token, insertion of a random token, or swap of two random tokens.

The network is tasked with generating the original sequence from the error injected one. This task better models real data than the sorting task above, because each generated token must follow the grammar and is therefore conditioned on all previous tokens. The results in Table 1 show that our proposed GAN approach is able to achieve high CFG accuracy, in terms of generating correct sequences that fit the CFG. In addition to CFG accuracy, we also compute BLEU scores based on the pairs before and after error injection. We should note that our random error injection process results in many bad examples corresponding to a specific good example or vice verse, i.e., mappings are not bijective. Having multiple bad examples in the dataset paired with the same good example contributes to the slightly lower BLEU scores, since the network can only map each bad input to a single output. This issue appears frequently in real world repair datasets, since code sequences can be repaired or broken

Table 1: Results on all experiments. Cur refers to experiments using curriculum learning, while Auto and Freq are those using $\mathcal{L}_{AUTO}$ and $\mathcal{L}_{FREQ}$, respectively. Sate4-P and Sate4-U denote paired and unpaired datasets, respectively.

| | Sorting | | CFG | | Sate4-P | Sate4-U |
|---|---|---|---|---|---|---|
| Model | Seq Acc. | Order Acc | BLEU-4 | CFG Acc | BLEU-4 | BLEU-4 |
| **seq2seq** | | | | | | |
| Base | **99.7** | **99.8** | **91.3** | **99.3** | 96.3 | N/A |
| Base + Cur | **99.7** | **99.8** | 90.2 | 98.9 | **96.4** | N/A |
| **Proposed GAN** | | | | | | |
| Base | 82.8 | 96.9 | 88.5 | 98.0 | 84.2 | 79.3 |
| Base + Auto | 98.9 | 99.6 | **88.6** | 96.5 | 85.7 | 79.2 |
| Base + Freq | **99.3** | **99.7** | 88.3 | 97.5 | 86.2 | 79.5 |
| Base + Cur | 81.5 | 98.0 | 88.4 | **98.9** | 88.3 | 81.1 |
| Base + Cur + Auto | 96.2 | 98.0 | 88.5 | 97.8 | 89.9 | **81.5** |
| Base + Cur + Freq | 98.2 | 99.1 | **88.6** | 96.3 | **90.3** | 81.3 |

multiple different ways. Our GAN approach performs well on this CFG dataset suggesting that it can handle this issue for which cycle approaches are not appropriate [28, 29, 27].

## 5.3   SATE IV

SATE IV is a dataset which contains C/C++ synthetic code examples (functions) with vulnerabilities from 116 different Common Weakness Enumeration (CWE) classes, and was originally designed to explore performance of static and dynamic analyzers [38]. Each bad function contains a specific vulnerability, and is paired with several candidate repairs. There is a total of $117,738$ functions of which $41,171$ contain a vulnerability and $76,567$ do not. We lex each function using our custom lexer. After lexing, each function ranges in length from 10 to 300 tokens.

Using this data, we created two datasets to perform two different experiments, namely *paired* and *unpaired* datasets. The paired dataset allows us to compare the performance of our GAN approach with a seq2seq approach. In order to have a dataset which is fair for both GAN and seq2seq training, we created paired data by taking each example of vulnerable code and sampling one of its repairs randomly. We iterate this process through the dataset four times, pairing each vulnerable function with a sampled repair, and combine the resulting sets into a single large dataset. We should mention that although the paired dataset includes labeled pairs, those labels are not utilized for GAN training. For the unpaired dataset, we wanted to guarantee that a given source sequence does not have a corresponding target sequence anywhere in the training data. To achieve this, we divided the data into two disjoint sets by placing either a vulnerable function or its candidate repairs into the training dataset with equal probability. Note that this operation reduces the size of our training data by half. For testing, we compute BLEU scores using all of the candidate repairs for each vulnerable function. We use a 80/10/10% train/validation/test split.

As shown in Table 1, our proposed GAN approach achieves progressively better results when we add (a) curriculum training, and (b) either $\mathcal{L}_{AUTO}$ or $\mathcal{L}_{FREQ}$ regularization loss. The Base + Cur + Freq model proves to be the best among different GAN models, and performs reasonably close to the seq2seq baseline, which is the upper performance bound. The results on the unpaired dataset are fairly close to those achieved by the paired dataset, particularly in the Base case, even though they are obtained with only half of the training data. Some code examples where our GAN makes correct repairs are provided in Table 2, with additional examples in the Supplementary Material.

## 6   Conclusions

We have proposed a GAN based approach to train an NMT system for discrete domain mapping applications. The major advantage of our approach is that it can be used in the absence of paired data, opening up a wide set of previously unusable data sets for the

Table 2: Successful Repairs: (Top) This function calls sprintf to print out two strings, but only provides the first string to print. Our GAN repairs it by providing a second string. (Bottom) This function uses a variable again after freeing it. Our GAN repairs it by removing the first free.

| With Vulnerability | Repaired |
|---|---|
| ```void CWE685_Function__Call__With__Incorrect__
      Number__Of__Arguments() {
  char dst[DST__SZ];
  sprintf(dst, "%s %s", SRC__STR);
  printLine(dst);
}``` | ```void CWE685_Function__Call__With__Incorrect__
      Number__Of__Arguments() {
  char dst[DST__SZ];
  sprintf(dst, "%s %s", SRC__STR, SRC__STR);
  printLine(dst);
}``` |
| ```void CWE415__Double__Free___malloc__free__
      struct__31() {
  twoints *data;
  data = NULL;
  data = (twoints *)malloc(100 * sizeof(twoints));
  free(data);
  {
    twoints *data_copy = data;
    twoints *data = data_copy;
    free(data);
  }
}``` | ```void CWE415__Double__Free___malloc__free__
      struct__31() {
  twoints *data;
  data = NULL;
  data = (twoints *)malloc(100 * sizeof(twoints));

  {
    twoints *data_copy = data;
    twoints *data = data_copy;
    free(data);
  }
}``` |

sequence correction task. Key to our approach is the addition of two novel generator loss functions which enforce accurate domain mapping without needing multiple generators or domains to be bijective. We also have discussed, and made some progress, toward handling discrete outputs with GANs. We note that this problem is far from solved, however, and will likely inspire more research. Even though we only apply our approach to the problem of source code correction, it is applicable to other sequence correction problems, such as Grammatical Error Correction or language sentiment translation, e.g., converting negative reviews into positive ones.

**Acknowledgments**

This project was sponsored by the Air Force Research Laboratory (AFRL) as part of the DARPA MUSE program.

## Footnotes

*Work done while author was affiliated with Draper.

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
