[Supplementary Material]

# Learning to Repair Software Vulnerabilities with Generative Adversarial Networks - Supplementary Material

**Jacob A. Harer**[1,2]**, Onur Ozdemir**[1]**, Tomo Lazovich**[3*]**, Christopher P. Reale**[1]**,
Rebecca L. Russell**[1]**, Louis Y. Kim**[1]**, Peter Chin**[2]

[1]Draper, Cambridge, MA
[2]Department of Computer Science, Boston University, Boston, MA
[3]Lightmatter, Boston, MA

{jharer,oozdemir,creale,rrussell,lkim}@draper.com,
tomo@lightmatter.ai, spchin@cs.bu.edu

## 1 Network and Training Details

Here we provide additional network and training details useful for experimental replication. All of the networks used in this paper use a similar architecture but vary in the number and size of layers.

### 1.1 Network Architecture

For all the experiments, we use identical networks for the Generator in our GAN model as well as in the NMT model in our seq2seq baseline. Thus when we refer to generator in the rest of the section it applies to both models. Our network architecture is shown in Figure 1.

Our generator consists of two RNNs, an encoder and a decoder. For Sorting and CFG experiments, the generator RNNs contain 3 layers of 512 neurons each. For Sate4, it contains 4 layers of 512 neurons each. The encoder RNN processes the input sequence and produces a set of hidden states $h_t$. The final hidden state $h_T$ is used as the initial state to the decoder RNN which generates outputs $s_t$ one at a time, feeding its outputs back as input to $t_1$ until an end of sequence character is produced. The decoder and encoder are linked using global dot product attention as per [1].

All networks share the same discriminator architecture. Discriminator inputs $(s_0, s_1...s_T)$ each in $\mathbb{R}^k$ are concatenated into a matrix $\mathbf{G}$ in $\mathbb{R}^{T \times k}$. They are then passed through a single 1D convolutional layer with 300 filters each of sizes of 3, 7, and 11. These outputs are then aggregated and fed into a max pooling operation over the entire sequence length. This is fed into two fully connected layers, the first with 512 neurons, and the second with a single neuron, the output of the discriminator.

### 1.2 Training

We first train our generator as a denoising autoencoder for which we use the Adam optimization algorithm with a learning rate of $10^{-4}$. The same pretrained network is used to initialize the generator for all GAN and seq2seq networks.

GAN networks are trained using the RMSprop optimization algorithm. Learning rates are initialized to $5 * 10^{-4}$ for the discriminator and $10^{-5}$ for the generator. We train the discriminator 15 times for every generator update. Seq2seq models are trained using the

Figure 1: (Left) Generator consisting of N encoder layers feeding N decoder layers. Outputs from the encoder are also used as inputs to the attention mechanism with the query coming from the decoder output. (Right) Discriminator consisting of N convolution layers, a temporal max pooling, and N fully connected layers.

Adam optimizer with a learning rate of $10^{-4}$. We experimented extensively with varying the learning rate but found that increasing the discriminator learning rate made it unstable causing its accuracy to decrease. Increasing the generator learning rate causes it to update to quickly for the discriminator, meaning the discriminator would not remain close to optimal and therefore gradients through it were not reliable. In order to ensure that the discriminator starts at a good initial point, we initialize it by training it alone for the first 10 epochs. The generators learning rate is decayed by a factor of 0.9 every 10 epochs. In models where we employ curriculum learning, this decay is only performed after the curriculum is completed. Networks are trained for 200, 400, and 1000 epochs for the sorting, CFG, and SATE4 experiments, respectively.

GAN training uses the original clipped version of Wasserstein GAN [2] with a clipping threshold of 0.05. We also experimented heavily with this threshold, and found that a lower threshold led to low discriminator accuracy, and a higher threshold led to the same discrete domain issues as discussed in the paper for WGAN-I in Section 3.2.

Our curriculum clips each sequence to a given length. We step up the curriculum length either when the discriminator accuracy falls below 55% or after 40 epochs, whichever comes first. Sorting and CFG curriculum starts at sequence length 5 and is increased by 2 at each step. SATE4 curriculum starts at length 75 and is increased by 5 at each step.

## 2 Context Free Grammar

Our simple CFG experiment uses the following CFG. Each line is a production rule with possible sequences separated by |. Symbols in quotes are terminals.

```
S: SOS NP VP EOS
SOS: '1'
EOS: '2'
NP: Det Nom | PropN
Nom: Adj N | N
VP: V NP | V NP PP
PP: P NP
PropN: '3' | '4' | '5'
Det: '6' | '7'
N: '8' | '9' | '10' | '11' | '12'
Adj : '13' | '14' |  '15' | '16' | '17'
V:  '18'  | '19' | '20' | '21'
P: '22' | '23'
```

## 3 Repair Examples

Here we provide additional selected examples of source code correctly and incorrectly repaired by our GAN model. Tables 1-4 show successfull repairs, and Tables 5-6 show failures.

Table 1: Successful Repair - This functions reads the index of an array access from a socket and returns the memory at the index. The vulnerable function only checks the lower bound on the array size. Our GAN repairs it by adding an additional check on the upper bound.

| With Vulnerability | Repaired |
| --- | --- |

With Vulnerability:

```
void CWE129_Improper_Validation_Of_
      Array_Index() {
  int data;
  data = −1;
  {
    ifdef __WIN32 WSADATA wsa_data;
    int wsa_data_init = 0;
    endif int recv_rv;
    struct sockaddr_in s_in;
    SOCKET connect_socket = INVALID_SOCKET;
    char input_buf[CHAR_ARRAY_SIZE];
    do {
      ifdef __WIN32 if (WSAStartup(MAKEWORD(2,
          2), &wsa_data) != NO_ERROR) break;
      wsa_data_init = 1;
      endif connect_socket = socket(AF_INET,
          SOCK_STREAM, IPPROTO_TCP);
      if (connect_socket == INVALID_SOCKET)
        break;
      memset(&s_in, 0, sizeof(s_in));
      s_in.sin_family = AF_INET;
      s_in.sin_addr.s_addr = inet_addr("127.0.0.1");
      s_in.sin_port = htons(TCP_PORT);
      if (connect(connect_socket, (struct sockaddr
          *)\&s_in, sizeof(s_in)) ==
        SOCKET_ERROR)
        break;
      recv_rv = recv(connect_socket, input_buf,
          CHAR_ARRAY_SIZE, 0);
      if (recv_rv == SOCKET_ERROR || recv_rv
          == 0)
        break;
      data = atoi(input_buf);
    } while (0);
    if (connect_socket != INVALID_SOCKET)
      CLOSE_SOCKET(connect_socket);
    ifdef __WIN32 if (wsa_data_init) WSACleanup();
    endif
  }
  {
    int data_copy = data;
    int data = data_copy;
    {
      int data_buf[10] = {0, 1, 2, 3, 4, 5, 6, 7, 8,
          9};
      if (data > = 0) {
        printIntLine(data_buf[data]);
      } else {
        printLine("ERROR: Array index is negative");
      }
    }
  }
}
```

Repaired:

```
void CWE129_Improper_Validation_Of_
      Array_Index() {
  int data;
  data = −1;
  {
    ifdef __WIN32 WSADATA wsa_data;
    int wsa_data_init = 0;
    endif int recv_rv;
    struct sockaddr_in s_in;
    SOCKET connect_socket = INVALID_SOCKET;
    char input_buf[CHAR_ARRAY_SIZE];
    do {
      ifdef __WIN32 if (WSAStartup(MAKEWORD(2,
          2), &wsa_data) != NO_ERROR) break;
      wsa_data_init = 1;
      endif connect_socket = socket(AF_INET,
          SOCK_STREAM, IPPROTO_TCP);
      if (connect_socket == INVALID_SOCKET)
        break;
      memset(&s_in, 0, sizeof(s_in));
      s_in.sin_family = AF_INET;
      s_in.sin_addr.s_addr = inet_addr("127.0.0.1");
      s_in.sin_port = htons(TCP_PORT);
      if (connect(connect_socket, (struct sockaddr
          *)&s_in, sizeof(s_in)) ==
        SOCKET_ERROR)
        break;
      recv_rv = recv(connect_socket, input_buf,
          CHAR_ARRAY_SIZE, 0);
      if (recv_rv == SOCKET_ERROR || recv_rv
          == 0)
        break;
      data = atoi(input_buf);
    } while (0);
    if (connect_socket != INVALID_SOCKET)
      CLOSE_SOCKET(connect_socket);
    ifdef __WIN32 if (wsa_data_init) WSACleanup();
    endif
  }
  {
    int data_copy = data;
    int data = data_copy;
    {
      int data_buf[10] = {0, 1, 2, 3, 4, 5, 6, 7, 8,
          9};
      if (data > = 0 && data < 10) {
        printIntLine(data_buf[data]);
      } else {
        printLine("ERROR: Array index is
            out−of−bounds");
      }
    }
  }
}
```

Table 2: Successful Repair - This function attempts to accept a socket and use it before it has bound it. Our GAN approach repairs the function by reordering the bind, listen, and accept into the correct order.

| With Vulnerability | Repaired |
|---|---|
| ```c
void CWE666_Operation_on_Resource_in_Wrong_
    Phase_of_Lifetime__accept_listen_bind__() {
  {
    char data[100] = "";
    ifdef __WIN32 WSADATA wsa_data;
    int wsa_data_init = 0;
    endif int recv_rv;
    struct sockaddr_in s_in;
    char *replace;
    SOCKET listen_socket = INVALID_SOCKET;
    SOCKET accept_socket = INVALID_SOCKET;
    size_t data_len = strlen(data);
    do {
      ifdef __WIN32 if (WSAStartup(MAKEWORD(2,
          2), &wsa_data) != NO_ERROR) break;
      wsa_data_init = 1;
      endif listen_socket = socket(AF_INET,
          SOCK_STREAM, IPPROTO_TCP);
      if (listen_socket == INVALID_SOCKET)
        break;
      memset(&s_in, 0, sizeof(s_in));
      s_in.sin_family = AF_INET;
      s_in.sin_addr.s_addr = INADDR_ANY;
      s_in.sin_port = htons(TCP_PORT);
      accept_socket = accept(listen_socket, NULL,
          NULL);
      if (accept_socket == SOCKET_ERROR)
        break;
      if ( listen ( listen \_socket,
          LISTEN\_BACKLOG)
          == SOCKET\_ERROR)
        break;
      if (bind(listen_socket, (struct sockaddr *)& s_in,
          sizeof(s_in)) == SOCKET_ERROR)
        break;
      recv_rv = recv(accept_socket, (char *)data +
          data_len,
                   (int)(100 − data_len − 1), 0);
      if (recv_rv == SOCKET_ERROR || recv_rv
          == 0)
        break;
      data[recv_rv] = '\0';
      replace = strchr(data, '\r');
      if (replace)
        *replace = '\0';
      replace = strchr(data, '\n');
      if (replace)
        *replace = '\0';
    } while (0);
    if (listen_socket != INVALID_SOCKET)
      CLOSE_SOCKET(listen_socket);
  }
}
``` | ```c
void CWE666_Operation_on_Resource_in_Wrong_
    Phase_of_Lifetime__accept_listen_bind__() {
  {
    char data[100] = "";
    ifdef __WIN32 WSADATA wsa_data;
    int wsa_data_init = 0;
    endif int recv_rv;
    struct sockaddr_in s_in;
    char *replace;
    SOCKET listen_socket = INVALID_SOCKET;
    SOCKET accept_socket = INVALID_SOCKET;
    size_t data_len = strlen(data);
    do {
      ifdef __WIN32 if (WSAStartup(MAKEWORD(2,
          2), &wsa_data) != NO_ERROR) break;
      wsa_data_init = 1;
      endif listen_socket = socket(AF_INET,
          SOCK_STREAM, IPPROTO_TCP);
      if (listen_socket == INVALID_SOCKET)
        break;
      memset(&s_in, 0, sizeof(s_in));
      s_in.sin_family = AF_INET;
      s_in.sin_addr.s_addr = INADDR_ANY;
      s_in.sin_port = htons(TCP_PORT);
      if (bind(listen_socket, (struct sockaddr*)&s_in,
          sizeof(s_in)) == SOCKET_ERROR)
        break;
      if ( listen (listen_socket, LISTEN_BACKLOG)
          == SOCKET\_ERROR)
        break;
      accept_socket = accept(listen_socket, NULL,
          NULL);
      if (accept_socket == SOCKET_ERROR)
        break;
      recv_rv = recv(accept_socket, (char *)data +
          data_len,
                   (int)(100 − data_len − 1), 0);
      if (recv_rv == SOCKET_ERROR || recv_rv
          == 0)
        break;
      data[recv_rv] = '\0';
      replace = strchr(data, '\r');
      if (replace)
        *replace = '\0';
      replace = strchr(data, '\n');
      if (replace)
        *replace = '\0';
    } while (0);
    if (listen_socket != INVALID_SOCKET)
      CLOSE_SOCKET(listen_socket);
  }
}
``` |

Table 3: Successful Repair - This function has a buffer allocated which is too small for the resulting data write. Our GAN repairs it by increasing the amount of memory allocated to the buffer.

| With Vulnerability | Repaired |
|---|---|
| ```c
void CWE131_Incorrect_Calculation_Of_
    Buffer_Size() {
  wchar_t *data;
  data = NULL;
  data = (wchar_t *)malloc(10 * sizeof(wchar_t));
  {
    wchar_t data_src[10 + 1] = SRC_STRING;
    size_t i, src_len;
    src_len = wcslen(data_src);
    for (i = 0; i < src_len; i++) {
      data[i] = data_src[i];
    }
    data[wcslen(data_src)] = L '\0';
    printWLine(data);
    free(data);
  }
}
``` | ```c
void CWE131_Incorrect_Calculation_Of_
    Buffer_Size() {
  wchar_t *data;
  data = NULL;
  data = (wchar_t *)malloc((10 + 1) *
      sizeof(wchar_t));
  {
    wchar_t data_src[10 + 1] = SRC_STRING;
    size_t i, src_len;
    src_len = wcslen(data_src);
    for (i = 0; i < src_len; i++) {
      data[i] = data_src[i];
    }
    data[wcslen(data_src)] = L '\0';
    printWLine(data);
    free(data);
  }
}
``` |

Table 4: Successful Repair - This function calls sprintf to print out two strings, but only provides the first string to print. Our GAN repairs it by providing a second string.

| With Vulnerability | Repaired |
|---|---|
| ```c
void CWE685_Function_Call_With_Incorrect_
    Number_Of_Arguments() {
  char dst[DST_SZ];
  sprintf(dst, "%s %s", SRC_STR);
  printLine(dst);
}
``` | ```c
void CWE685_Function_Call_With_Incorrect_
    Number_Of_Arguments() {
  char dst[DST_SZ];
  sprintf(dst, "%s %s", SRC_STR, SRC_STR);
  printLine(dst);
}
``` |

Table 5: Incorrect Repair - This function tries to print out from data which has not been fully initialized. Our GAN corrects this by fully initializing the variable, but then attempts to call a variable identifier instead of making the appropriate function call.

| With Vulnerability | GAN Attempted Repaired | Golden Repair |
|---|---|---|
| ```c
void CWE457_Use_of
    _Uninitialized_Variable() {
  double *data;
  data = (double *)ALLOCA(10 *
      sizeof(double));
  {
    int i;
    for (i = 0; i < (10 / 2); i++)
        {
      data[i] = (double)i;
    }
  }
  {
    int i;
    for (i = 0; i < 10; i++) {
      printDoubleLine(data[i]);
    }
  }
}
``` | ```c
void CWE457_Use_of_
    Uninitialized_Variable(){
  double *data;
  data = (double *)ALLOCA(10 *
      sizeof(double));
  {
    size_t i;
    for (i = 0; i < 10; i++) {
      data[i] = (double)i;
    }
  }
  {
    int i;
    for (i = 0; i < 10; i++) {
      static_five(data[i]);
    }
  }
}
``` | ```c
void CWE457_Use_of_
    Uninitialized_Variable() {
  double *data;
  data = (double *)ALLOCA(10 *
      sizeof(double));
  {
    int i;
    for (i = 0; i < 10; i++) {
      data[i] = (double)i;
    }
  }
  {
    int i;
    for (i = 0; i < 10; i++) {
      printDoubleLine(data[i]);
    }
  }
}
``` |

Table 6: Incorrect Repair - This function encounters an integer underflow by assigning a random value to a char and then subtracting 1. If the random value is 0 this will underflow the char. The given golden repair in this case is simply to change the random char to a known value. However, our GAN gets confused. It instead modifies the rand function in an unknown way and proceeds to free the data rather then print it.

| With Vulnerability | GAN Attempted Repaired | Golden Repair |
|---|---|---|
| ```c
void CWE191_Integer_
    Underflow() {
  unsigned char data;
  data = ' ';
  data = (unsigned char)rand();

  {
    unsigned char result =
        data - 1;
    printHexUnsignedCharLine(result);
  }
}
``` | ```c
void CWE191_Integer_
    Underflow(){
  unsigned char data;
  data = ' ';
  data = (unsigned char)
      rand((unsigned int)data);

  {
    char data = data;

    free(data);
  }
}
``` | ```c
void CWE191_Integer_
    Underflow() {
  unsigned char data;
  data = ' ';
  data = 5;

  {
    unsigned char result =
        data - 1;
    printHexUnsignedCharLine(result);
  }
}
``` |

## Footnotes

*Work done while author was affiliated with Draper.