[Reviews · NeurIPS 2018]

Reviewer 1



Update based on author rebuttal: The authors address some of my criticisms and promise to improve some of the motivation in subsequent drafts. The rebuttal doesn't change my impression of accepting this paper. This paper proposes a system for correcting sequences, with a target application of fixing buggy source code. They use a sequence-to-sequence model within a GAN framework, which allows the model to be trained without paired source/target data. Some additional new tricks are proposed to make the model output consistent translations of the input. The model is tested on two synthetic tasks and a source code correction benchmark. Overall, I thought this paper provided a compelling model which seems to function well on sequence-to-sequence tasks even without explicit source/target pairs. The authors focus only on datasets where explicit source/target pairs are available in order to compare their method to standard seq2seq models. This was a disappointment to me. As the authors say, "The major advantage of our approach is that it can be used in the absence of paired data, opening up a wide set of previously unusable data sets for the sequence correction task." While the experiments suggest the model can still perform reasonably well without explicitly using the source-target pairs, the model itself was not tested in this regime. In particular, a truly realistic test of the model would be in a setting where there is no guarantee that a given source sequence has a paired target sequence anywhere in the training data. I think this paper would be a lot stronger if it was shown to work in this setting - perhaps by being deployed in a source code vulnerability detection system and proving to be genuinely useful. That being said, I am not familiar with the source code correction problem domain so I may be missing the significance of the reported results. Apart from this, I found some of the presentation confusing or otherwise amenable to improvement; some specific comments are below. > "only a limited number of relatively small paired sequence correction datasets exist" In the context (and via the citation), it sounds like you are discussing machine translation datasets, not "sequence correction" datasets, but in that case this statement is false - there are many translation datasets, and many of them are large. From later context, I think you mean generic datasets which deal with correcting sequences (like fixing grammar errors, or source code correction), but I was thrown off by this sentence in the midst of a paragraph about MT. - In your description of WGAN, you don't mention that in order to actually estimate Wasserstein distance the function class must be constrained to 1-Lipschitz functions, for example by weight clipping (as originally proposed in WGAN) or via a gradient penalty (as proposed by Gulrajani et al.) You mention this later, but it's inaccurate to introduce the WGAN the way that you do without mentioning this. - You state that you feed the discriminator "soft" network outputs directly rather than sampling. Do you still sample from the generator's output and feed the sample back in? If you don't, your model is not very powerful because it no longer has a stochastic variable and all possible future outcomes must be encoded in the deterministic hidden state. If you do, then you can't propagate gradients though the sampling operation so your model is only learning one-step dependencies. Which did you do and why? From later discussion (4.2) it sounds like the latter; I'd suggest addressing this early on. - Your discussion of the weak discriminator feels a little incomplete to me. It is nice that with a weak discriminator you obtain useful gradients, but there is a trade-off between discriminator strength and the generator's abilities, namely because the generator only ever learns how to solve the task through the discriminator. Being able to utilize discriminator gradients is not enough - the discriminator has to be able to provide a good model of the data distribution in order to be useful. Of course, as you point out, useful gradients are a prerequisite, but my point is that the generator's performance is limited in the case of a weak generator. I would suggest adding a few sentences to comment on this. - More common is to call the "Improved WGAN" loss the "WGAN-GP" loss, due to the gradient penalty.

Reviewer 2



This paper proposes an adversarial learning approach based on GAN for repairing software vulnerabilities. GAN learns the mapping from one discrete source domain to another target domain, which does not require labeled pairs. The proposed model is shown effectiveness and the experimental performance is close to previous seq2seq approach using labeled pairs of data. Pros: * Security vulnerabilities in software programs pose serious risks and automatically repairing vulnerabilities is an important task to study. * It is much easier to obtain unpaired data in many seq2seq applications. The idea of using GAN to learn a mapping between two domains is novel, which is effective to train the model without paired examples. * The paper is well-organized and easy to follow. The proposed GAN framework and the two regularizations are generally sound. * The experimental results prove the effectiveness of GAN for vulnerabilities repairing, which obtains comparable performance of state-of-the-art seq2seq models which use labeled pairs of data. Cons: * The discussion of the differences in the results of AUTO and PREQ regularization could be added in the experiment section.

Reviewer 3



The authors explore and develop an approach to code correction, or repairing software vulnerabilities, (in general sequence correction) via a (Wasserstein) GAN and Neural Model Translation (NMT) approach, that does not require having paired examples for training. The approach only requires having ample examples of each domain (faulty codes and correct codes, or more generally, good and bad sentences/sequences). The authors have to overcome a number of challenges, and show that they get favorable accuracy compared to seq2seq techniques. The experiments are performed on two synthetic/controlled domains, and one with C++ and software vulnerabilities (10s of thousands of functions). Recent work on NMT with monolingual corpora and unsupervised cypher cracking are perhaps closest in nature to this work, but the author explain a major difference. The authors explore a two regularizations and as well as curriculum learning approach to (potentially) improve results. I didn't get an idea of how far from practical utility, in fixing vulnerabilities, the approach is.. even for seq2seq which has a better accuracy of 96% (vs 90 for proposed GAN). For instance, do the approaches output good/reliable probabilities, and can one achieve reasonable recall with say 99% confidence? It would help to further motivate the problem. It appears that even detecting a (nontrivial) vulnerability is a challenging task, and manually fixing such (vs automated) may not be the main problem. Update after author rebuttal: The authors promise to improve the motivation & address questions raised. The rebuttal hasn't changed my review.